# Plasma-Derived miRNA-222 as a Candidate Marker for Papillary Thyroid Cancer

**DOI:** 10.3390/ijms21176445

**Published:** 2020-09-03

**Authors:** Aistė Kondrotienė, Albertas Daukša, Daina Pamedytytė, Mintautė Kazokaitė, Aurelija Žvirblienė, Dalia Daukšienė, Vaida Simanavičienė, Raimonda Klimaitė, Ieva Golubickaitė, Rytis Stakaitis, Valdas Šarauskas, Rasa Verkauskienė, Birutė Žilaitienė

**Affiliations:** 1Institute of Endocrinology, Medical Academy, Lithuanian University of Health Sciences, LT-50161 Kaunas, Lithuania; aiste.kondrotiene@lsmuni.lt (A.K.); mintaute.kazokaite@lsmuni.lt (M.K.); dalia.dauksiene@lsmu.lt (D.D.); raimonda.klimaite@lsmuni.lt (R.K.); rasa.verkauskiene@lsmuni.lt (R.V.); 2Institute of Digestive Research, Medical Academy, Faculty of Medicine, Lithuanian University of Health Sciences, LT-50161 Kaunas, Lithuania; albertas.dauksa@lsmuni.lt; 3Institute of Biotechnology, Life Sciences Center, Vilnius University, LT-10257 Vilnius, Lithuania; daina.pamedytyte@gmc.vu.lt (D.P.); aurelija.zvirbliene@bti.vu.lt (A.Ž.); vaida.simanav@gmail.com (V.S.); 4Institute of Biology systems and genetic research, Lithuanian University of Health Sciences, LT-50161 Kaunas, Lithuania; ieva.golubickaite@lsmuni.lt; 5Laboratory of Molecular Neurooncology, Neuroscience Institute, Lithuanian University of Health Sciences, LT-50161 Kaunas, Lithuania; rytis.stakaitis@lsmuni.lt; 6Department of Pathology, Lithuanian University of Health Sciences, LT-50161 Kaunas, Lithuania; valdas.sarauskas@lsmuni.lt

**Keywords:** papillary thyroid carcinoma, miR-146b, miR-222, miR-221, miR-21, miR-181b, plasma

## Abstract

We analyzed five miRNA molecules (miR-221; miR-222; miR-146b; miR-21; miR-181b) in the plasma of patients with papillary thyroid cancer (PTC), nodular goiter (NG) and healthy controls (HC) and evaluated their diagnostic value for differentiation of PTC from NG and HC. Preoperative PTC plasma miRNA expression (*n* = 49) was compared with plasma miRNA in the HC group (*n* = 57) and patients with NG (*n* = 23). It was demonstrated that miR-221; miR-222; miR-146b; miR-21 and miR-181b were overexpressed in preoperative PTC plasma samples compared to HC (*p* < 0.0001; *p* < 0.0001; *p* < 0.0001; *p* < 0.0001; *p* < 0.002; respectively). The upregulation in tumor tissue of these miRNAs was consistent with The Cancer Genome Atlas Thyroid Carcinoma dataset. A significant decrease in miR-21; miR-221; miR-146b and miR-181b expression was observed in the plasma of PTC patients after total thyroidectomy (*p* = 0.004; *p* = 0.001; *p* = 0.03; *p* = 0.036; respectively). The levels of miR-222 were significantly higher in the preoperative PTC compared to the NG group (*p* = 0.004). ROC curve (receiver operating characteristic curve) analysis revealed miR-222 as a potential marker in distinguishing PTC from NG (AUC 0.711; *p* = 0.004). In conclusion; circulating miR-222 profiles might be useful in discriminating PTC from NG.

## 1. Introduction

Papillary thyroid carcinoma (PTC) is the most common type of thyroid cancer (85%–90%) [1]. The 10-year survival rate in PTC after treatment is higher than 90% [2]. However, regional or distant metastatic recurrences occur in up to 10% of cases [3]. The current clinical difficulty is in properly identifying PTC cancer. Ultrasound diagnosis depends on the experience and knowledge of the examiner. Fine needle aspiration biopsy (FNAB) also has limitations, as it is quite challenging to take a biopsy from a small nodule [4]. Furthermore, FNAB diagnosis is doubtful in up to 20% of cases and then thyroid surgery is still needed to confirm malignancy [5].

It is important to differentiate PTC from benign thyroid nodule early enough to avoid an advanced disease course. Therefore, non-invasive biomarkers of PTC are needed [6]. The European Thyroid Association recommends investigating *BRAF, RET/PTC, PAX8/PPARG, RAS* mutations, *TERT* promotor and *TP53* in case of uncertainty about further clinical action depending on Bethesda classes [6]. Further miRNA investigations are encouraged in the same guidelines, as these molecules demonstrate a potential value in PTC diagnostics and follow up [6].

miRNAs are endogenous non-coding RNA molecules (19–25 nucleotides in length) identified as post-transcriptional negative regulators of gene expression by attaching to the 3′ untranslated region of target mRNAs in the cytoplasm [7,8]. miRNAs have been shown to be involved in fundamental biological processes, including metabolism, cell life cycles, tissue differentiation, embryogenesis and organogenesis functions such as cell proliferation and apoptosis, proving their relevance as regulators for oncogenes and tumor suppressor genes [7,9]. Extensive efforts have been made in determining miRNA expression changes in different diseases and cancer types [10]. Tissue and bioliquid (e.g., plasma, serum) miRNA is highly stable [11]. The fact that specific miRNAs are released from tissues into the circulation with the development of disease encourages further studies on the detection of non-invasive biomarkers [11]. Functions of circulating miRNAs are being investigated and hormone-like activity of circulating miRNAs at long distances has been recognized in recent years [12]. However, identification of specific roles of circulating miRNAs, understanding their implication in specific pathways and finding the most sensitive and specific miRNAs is still challenging [13].

It is already known that miR-146b, miR-21, miR-221, miR-222, miR-181b are dysregulated in PTC. These molecules showed potential as biomarkers in PTC tissue and plasma/serum studies [14,15,16,17,18,19,20,21,22,23]. It was also demonstrated that downregulation of miR-181b inhibits proliferation and promotes apoptosis in thyroid cancer TPC1 cells, making this miRNA a possible target for PTC treatment [24]. Moreover, miR-181b was also found to be upregulated in PTC tissue samples [25]. miR-21 was found to be overexpressed in PTC compared to healthy thyroid tissue [20], as well as in recurrent PTC compared to non-recurrent PTC [18,20]. This miRNA has diagnostic value in head and neck cancer as it is released into the bloodstream [26]. miRNA-21 promotes cell proliferation and invasion via the VHL/PI3K/AKT pathway [27]. miRNA-222 and miRNA-221 also have potential to become biomarkers of PTC. The putative targets for miRNA-222 and -221 are the p27/kip1 and p57/kip2 genes, which affect cell cycle regulation and cancer inhibition [28]. Increased expression of these miRNAs results in downregulation of these genes, which reduces genome instability and cell proliferation [28]. The many effects of miR-146b seen in PTC cells may indicate the significance of this regulatory molecule in the diagnosis and prognosis of PTC [29]. miR-146b was found to increase cell proliferation activity and inhibit cell cycle arrest by downregulating SMAD4 [30]. Moreover, it promotes migration, invasion and epithelial-to-mesenchymal transition by downregulating ZNRF3 [31] and increases cell proliferation by downregulating IRAK1 [32].

The aim of our study was to analyze the expression of five miRNA molecules (miR-21; miR-221; miR-222; miR-146b; miR-181b) in plasma samples and evaluate the differences in patients with PTC, benign nodules and healthy controls to identify the potential biomarkers of PTC for non-invasive diagnostics.

## 2. Results

### 2.1. Characteristics of the Study Groups

Forty-nine patients with papillary thyroid carcinoma (PTC), 23 with nodular goiter (NG) and 57 healthy controls (HC) participated in the study. The mean age of PTC patients was 47.86 ± 12.1, that of NG patients was 50.09 ± 19.3 and that of HC was 45.43 ± 12.98 (*p* > 0.05).

In the PTC group, there were 8 male patients (16.32%) and 41 female patients (83.68%), while in the NG group, 1 patient (4.2%) was male and 22 patients (95.7%) were female and in the HC group, 13 participants (22.8%) were male and 44 participants (77.19%) were female (*p* > 0.05).

Among the PTC patients, 8 (16.32%) had the classical variant of PTC, 7 (14.29%) had the follicular variant of PTC, 12 (24.48%) had the diffuse sclerosing variant of PTC and 22 (44.89%) had microcarcinoma. PTC tumor sizes were distributed as follows: 19 (38.87%)—T1a, 5 (10.2%)—T1b, 3 (6.12%)—T2, 22 (44.89%)—T3. Twelve (24.49%) participants with PTC had lymph node metastases. Multifocality was observed in 14 (28.57%), infiltrative growth in 35 (71.42%), capsule overgrowth in 20 (40.81%), soft tissue infiltration in 8 (16.32%) and lymphovascular invasion in 22 (44.89%) of the PTC patients.

### 2.2. miRNA Expression Analysis from the The Cancer Genome Atlas (TCGA) Database

Webtool miR-TV [33] was used for the evaluation of miRNA expression from the TCGA-Thyroid cancer (THCA) project data. Normal vs. tumor sample comparison showed significant upregulation of miR-146b, miR-222, miR-221, miR-21 and miR-181b in primary tumors (Figure 1).

miRNA-Seq data obtained from the TCGA-THCA repository with additional filters for disease type, prior malignancy/treatment, race and ethnicity were used for a more accurate analysis. Only patients diagnosed with adenocarcinoma without prior malignancy or treatment were included. To obtain data resembling the Lithuanian population, white race with not Hispanic or Latino ethnicity was selected, resulting in 186 primary tumors and 31 solid normal tissue sample data. The significant upregulation pattern of miR-146b, miR-222, miR-221, miR-21 and miR-181b in primary tumors remain clear after additional data filtration (Figure 2).

To get the results that represent our study cohort even more, we selected to only analyze patients with data for both tumor and normal tissue adjacent to the tumor, resulting in 31 paired samples in each group. The upregulation of analyzed microRNAs in primary tumors remained consistent with the prior TCGA data analysis (Figure 3).

### 2.3. Plasma miRNA Levels in PTC, NG and HC Groups

miR-221, miR-222, miR-146b, miR-181b and miR-21 expression was significantly higher in the preoperative PTC group compared to the HC group (*p* < 0.0001, *p* = 0.002, *p* < 0.001, *p* < 0.001, *p* < 0.001, respectively) (Appendix A; Table A1). Only miRNA-222 expression was significantly higher in PTC patients compared to the NG group (*p* = 0.004). The expression of other investigated miRNAs did not differ significantly between PTC and NG groups (Table A2). Expression of miR-221, miR-21, miR-146b and miR-181b was significantly higher in the NG group compared to the HC (*p* = 0.02, *p* = 0.008, *p* = 0.033 and *p* = 0.003, respectively) (Figure 4; Table A3).

### 2.4. Plasma miRNA Expression in PTC Patients Before and After Surgery

Plasma miR-146b, miR-222, miR-21, miR-221 and miR-181b expression after surgery was evaluated depending on the extent of surgery (hemi thyroidectomy or total thyroidectomy). In a group of PTC patients who underwent total thyroidectomy, the expression of miR-146b, miR-21, miR-221 and miR-181b in plasma was significantly lower after surgery (*p* = 0.03; 0.004; 0.001; 0.038, respectively) (Table 1). Only miR-221 showed significantly lower plasma expression after hemi thyroidectomy (Table 2).

### 2.5. Plasma miRNA Expression in Relation to the Clinicopathological PTC Features

We analyzed plasma miRNA expression in relation to the clinicopathological PTC features. Plasma expression of miR-222 was found to be higher in multifocal PTC compared to unifocal PTC (*p* = 0.024) (Table 3). The tendency of miR-181b overexpression in PTC with lymphovascular invasion was observed but no statistically significant relation was found (*p* = 0.076).

To evaluate the diagnostic value of circulating miR-146b, miR-222, miR-21, miR-221 and miR-181b, ROC (receiver operating characteristic) curve analysis was performed. All five miRNAs had statistically significant satisfactory or good diagnostic values to differentiate PTC patients from healthy controls (Figure 5A,C,E,G,J; Appendix B; Table A4 and Table A5). miR-221 had the highest AUC of 0.792 (95% CI = 0.708–0.877), with 73.5% sensitivity and 73.7% specificity at the cutoff value of 0.0104 (Figure 5G).

The expression of miR-21, miR-221, miR-146b and miR-181b did not show statistically significant differences between PTC from NG (Figure 5B, 5F, 5H, 5K; Appendix B; Table A4 and Table A5). We calculated expression of miR-221, miR-21, miR-222, miR-146b and miR-181b as one singular entity for ROC analysis. The AUC was 0.579 (95% CI = 0.518–0.639) (*p* = 0.02) in discriminating PTC from NG and the AUC was 0.549 (95% CI = 0.504–0.594) (*p* = 0.034) in discriminating PTC from HC, with no differences seen between the five miRNAs panel.

However, the comparison between PTC patients and those with benign nodules showed that plasma expression of miR-222 had an AUC of 0.711 (95% CI = 0.587–0.834), with 61.2% sensitivity and 78.3% specificity at the cutoff value of 0.032 (*p* < 0.004). This might be useful for differentiating PTC from NG (Figure 5D).

### 2.6. Analysis of miR-222 Targets

Target prediction analysis indicated four major miR-222 mRNA targets—*ARF4* (ADP Ribosylation Factor 4), *DCAF12* (DDB1 And CUL4 Associated Factor 12), *CDKN1B* (Cyclin Dependent Kinase Inhibitor 1B), *MYLIP* (Myosin Regulatory Light Chain Interacting Protein). The expression of these targets was decreased in tumor tissue, suggesting the silencing function of miR-222 to these genes (Figure 6).

## 3. Discussion

In this study, we investigated five miRNAs (miR-146b, miR-21, miR-221, miR-222, miR-181b) as possible non-invasive biomarkers for PTC. We compared their expression profiles in plasma samples of PTC, NG and HC study groups and analyzed their relation to the clinicopathologic characteristics of PTC. We determined a significant overexpression of miR-146, miR-221, miR-222, miR-21 and miR-181b in PTC plasma samples compared to HC. However, only miR-222 showed a significant difference between PTC and NG study groups. In addition, miR-222 expression was associated with PTC multifocality. ROC analyses indicated that all five miRNAs had statistically significant expression changes that could differentiate PTC patients from healthy controls. Moreover, ROC analysis revealed that miR-222 is a possible plasma-derived diagnostic biomarker for distinguishing PTC from NG.

Tumor-derived miRNAs are released into the circulation [34]; therefore, specific types of circulating miRNAs from each organ may have a diagnostic and prognostic role in different types of cancer [35,36,37,38]. Circulating miRNAs have been demonstrated to have an impact on cell–cell communication in tumor biology [39]. The expression of circulating miRNAs is usually evaluated in the plasma or serum [11]. Coagulation may affect the spectrum of extracellular miRNA in the blood [40,41]. We chose to analyze miRNA expression in plasma samples because this bioliquid is prevented from clotting [40,41]. Certain studies have already shown changes in circulating miRNA expression in PTC patients [42,43,44]. Our findings suggest that these five miRNAs are overexpressed in PTC and spread from PTC cells to the bloodstream at detectable levels.

Previous studies have demonstrated that miR-146b, miR-221 and miR-222 are the most consistently overexpressed miRNAs in PTC tissue [14,22,45,46]. Moreover, studies that explored miR-146b, miR-222 and miR-221 expression in plasma samples showed their consistent upregulation in PTC compared to the HC group [22,40]. To our best knowledge, miR-181b and miR-21 have not been investigated in plasma as possible biomarkers of PTC, although they have been found to be overexpressed in PTC tissue samples [14,18,25]. Furthermore, miR-21 was analyzed in serum samples and showed potential to become a diagnostic and prognostic marker of PTC [42,43]. We found a significant overexpression of plasma miR-222 in PTC compared to NG. Previous studies also revealed an increase in miR-222 expression in PTC tissue samples as well as plasma samples. M.Rezaei et al. [44] analyzed miR-222, miR-146a and miR-181a in PTC and NG plasma samples and demonstrated an overexpression of these three miRNAs in the PTC group. Higher expression levels of miR-222 were also observed in PTC patient compared to NG patient serum samples [16]. However, caution must be taken when comparing miRNA data obtained from different sample types [47]. Our findings show that plasma expression of miR-221, miR-21, miR-146b and miR-181b was not statistically significant in PTC compared to NG, although previous studies have shown the potential diagnostic value of miR-146b [40].

If surgery is chosen for patients with thyroid cancer < 1 cm without extrathyroidal extension and cN0, the initial surgical procedure should be a thyroid lobectomy, unless there are clear indications to remove the contralateral lobe. For patients with thyroid cancer > 1 cm and < 4 cm without extrathyroidal extension and without clinical evidence of any lymph node metastases (cN0), the initial surgical procedure can be either a bilateral procedure (near total or total thyroidectomy) or a unilateral procedure (lobectomy). Thyroid lobectomy alone may be sufficient initial treatment for low-risk papillary and follicular carcinomas. For patients with thyroid cancer > 4 cm or with gross extrathyroidal extension (clinical T4) or clinically apparent metastatic disease in nodes (clinical N1) or distant sites (clinical M1), the initial surgical procedure should include a near-total or total thyroidectomy [48]. In our study, plasma levels of five miRNAs were measured in 37 PTC patients before and after surgery (30 patients underwent total thyroidectomy, 7 underwent hemi thyroidectomy). We checked if plasma miRNA expression also decreased in the absence of the tumor. The expression levels of miR-221, miR-21, miR-181b and miR-146b were significantly lower after total thyroidectomy compared with the samples before surgery. This suggests that miR-221, miR-21, miR-181b and miR-146b may have prognostic potential in PTC. Only miR-21 showed a significant reduction in its plasma levels after hemi thyroidectomy in PTC patients. This indicates that the prognostic potential of miR-221, miR-222, miR-146b and miR-181b after hemi thyroidectomy is doubtful since no significant decline is observed 4–6 weeks after surgery. Our observations differ from other studies that investigated the levels of specific circulating miRNAs as a marker to monitor the postoperative PTC progression. Reductions of 2.7–fold and 5.1–fold were observed in the plasma levels of miR-222 and miR-146b, respectively, after total thyroidectomy [22]. Zhang et al. evaluated the levels of miR-222, miR-221 and miR-146b via subsequent RT-qPCR during varied postoperative periods in the same patients [16]. The levels of miR-222, miR-221 and miR-146b rapidly decreased 1 month after surgery compared with their preoperative levels in the PTC group. There was no difference in the miR-222, miR-221 and miR-146b expression levels for patients with PTC undergoing hemi thyroidectomy or total thyroidectomy prior and after surgery [16]. Further studies of plasma miRNA expression after surgery with more subjects and a longer follow-up period would reveal prognostic value of these miRNAs to PTC patients.

As there is a clear association between miRNA expression and the clinicopathological features of PTC [20,49,50,51,52], our study revealed higher expression of miR-222 in plasma samples of patients with multifocal PTC. In a previous investigation of PTC tissues, associations of miR-221 [16], miR-146b [53], miR-146a [48] overexpression with PTC multifocality were found. Zhang Y. et al. analyzed the levels of circulating miR-222 in relation to PTC multifocality but no significant differences were found [16].

In our study, ROC curves were used to evaluate the diagnostic value of differently expressed miRNAs in PTC, HC and NG groups, which indicated that all five miRNAs showed statistically significant plasma expression changes that were able to differentiate PTC patients from healthy controls. Moreover, ROC curve analysis confirmed that plasma miR-222 might be a reliable marker in discriminating PTC from NG (AUC = 0.711 (95%CI 0.587–0.834), *p* = 0.004). ROC analysis was not used in previous studies that analyzed miRNA expression levels in the plasma of PTC, NG and/or HC. However, plasma miR-146b (AUC = 0.649 (95%CI 0.521–1.77)) was shown to be helpful to discriminate benign thyroid lesions from PTC in a study by Lee at al. [40]. Zhang Y et al. [16] proposed using circulating serum miR-222, miR-221 and miR-146b expression as a possible panel for distinguishing PTC from NG (AUC = 0.903 (95%CI 0.85–0.955)). However, the cohorts of these studies were small and therefore further investigations with a larger sample size are required.

For the first time we provided information about plasma expression of miRNA-21 and miRNA-181b in PTC patients. Furthermore, miRNA-222 plasma expression was found to be a promising diagnostic marker of PTC. Moreover, our findings suggest that plasma miRNA expression does not change significantly after hemi thyroidectomy and it therefore might be a useful prognostic marker after total thyroidectomy only. It should be noted that we evaluated expression of plasma miRNAs 4–6 weeks after surgery. Longer observational studies might reveal different results. Similar to previous studies, a limitation of our study is the small sample size. Therefore, it is difficult to make firm conclusions and further investigations are needed to confirm our results.

Target prediction analysis indicated four major miR-222 mRNA targets—*ARF4*, *DCAF12*, *CDKN1B*, *MYLIP*. Downregulation of *ARF4* in thyroid cancer results in inefficiency to accumulate radioiodine due to disrupted trafficking of sodium iodide symporter to the plasma membrane [54]. Interestingly, *ARF4* is described as an oncogene by promoting breast cancer cell migration and metastasis to the lungs [55,56]. Also, *ARF4* is reported to be upregulated in Epithelial ovarian cancer as well as in other major cancer tissues [57,58]. The combined downregulation of *ARF4* and upregulation of miR-222 might indicate its diagnostic specificity in thyroid cancer. *DCAF12* is mostly upregulated in various cancers tissues compared to normal ones and only in the adrenal gland, bone, testis and thyroid cancers it is downregulated [58]. This indicates the diagnostic potential of *DCAF12* and miR-222 joint expression evaluation in thyroid cancer. *CDKN1B* is associated with cell cycle regulation by regulating its division. Disruption of *CDKN1B* has been reported to evoke tumorigenesis by increased cell proliferation and loss of function [59]. The downregulation of *MYLIP* is associated with increased tumor migration and metastasis in breast cancer cells [60]. Also, the inhibition of *MYLIP* downregulation by *TUSC8* resulted in suppressed metastasis showing the important role of *MYLIP* in cancer [61].

The upregulation of miR-146b, miR-222, miR-221, miR-21 and miR-181b were consistent in both the TCGA-THCA dataset and our research cohort. These findings suggest an oncogenic role of these miRNAs in thyroid cancer. As miR-222, miR-181b, miR-146b, miR-21 and miR-222 are overexpressed in PTC tissue, our findings suggest that they circulate in a highly stable, cell-free form in plasma. Our study demonstrated overexpression of these five plasma miRNAs in the PTC group compared to healthy controls, while only miR-222 showed a significant difference between PTC and NG groups and an association with multifocality. Further studies with larger sample sizes are necessary to confirm its relevance as a biomarker for non-invasive diagnostics and prognosis of PTC.

## 4. Materials and Methods

### 4.1. Patient Groups

Patients with PTC, NG and HC were involved in this study. Plasma samples were obtained from patients with PTC (*n* = 49) who underwent total or hemi thyroidectomy one day before and 4–6 weeks after surgery at the Hospital of Lithuanian University of Health Sciences Kaunas Clinics between 2016 and 2018. Plasma samples from patients with NG were obtained one day before total or hemi thyroidectomy. The HC group (*n* = 57) had no thyroid disease or autoimmune or oncological illness and their family history for thyroid diseases was negative.

PTC and NG patients underwent thyroidectomy and the diagnosis was confirmed histopathologically after surgery. Classification of patients with PTC according to the 8th edition of the AJCC/UICC staging system was used.

The study was approved by the Kaunas Regional Committee of Biomedical Research (Lithuania, approval No. BE-2-44; 2015-12-23). Written informed consent was obtained from each participant of the study after full explanation of the purpose and nature of all procedures used. This study was conducted in accordance with the Declaration of Helsinki.

### 4.2. TCGA Database Analysis

Webtool miR-TV (Taipei, Taiwan) [33] was used for the primary evaluation of miRNAs’and their mRNA targets’ expression (targetScan version 7.2; miRDB version 5.0; miRanda, August 2010 release) from the TCGA-Thyroid cancer (THCA) project data (TCGA version 18.0). For filtered data evaluation, the newest version of TCGA dataset was used (version 25.0). Comparison performed using the Statannot package (version 0.2.3; https://github.com/webermarcolivier/statannot) with Python3 (version 3.7.3; Amsterdam, The Netherlands).

### 4.3. Plasma Samples

Venous blood was drawn from PTC patients one day before surgery and one month after the surgery and one day before surgery from NG patients. The samples of peripheral blood (10 mL) were collected into EDTA (BD Vacutainer PPT™ Plasma Preparation Tube; 13 × 100 mm/5 mL) venipuncture tubes. The blood was then centrifuged at 1900× *g* for 10 min at 4 °C. The plasma phase then was transferred to a new tube and centrifuged in conical tubes at 16,000× *g* for 10 min at 4 °C. The supernatant was transferred to 1.5 mL aliquots and stored at −80 °C until nucleic acid purification.

### 4.4. RNA Extraction

The miRNA was extracted from 200 μL of thawed plasma using a miRNeasy Serum/Plasma Kit (Qiagen, Hilden, Germany) according to the manufacturer’s protocol. The *Caenorhabditis elegans* miRNA-39 (spike-in cel-miR-39-3p) (Qiagen, Hilden, Germany) was used as a synthetic spike-in control for normalization. An equal amount (8 × 10^9^ copies) of *C. elegans* miR-39-3p was added to each serum sample before RNA isolation. The level of hemolysis in the plasma samples was assessed before miRNA extraction. The plasma (100 μL) was centrifuged at 1600× *g* for 4 min at 4 °C. Oxy-hemoglobin absorbance was measured at λ = 414 nm wavelength using a NanoDrop ND1000 Spectrophotometer (ThermoFisher Scientific, Waltham, MA, USA). The procedure was repeated 3–5 times and the average optical density (OD) was calculated. Plasma samples with OD_414_ > 0.25 were disqualified from further analysis.

### 4.5. Quantitative Reverse Transcription-Polymerase Chain Reaction

The expression levels of the miRNA were measured by a quantitative reverse transcription polymerase chain reaction (qRT-PCR) using a TaqMan Small RNA Assay (Applied Biosystems, Foster City, CA, USA) according to the manufacturer’s protocol. First complementary DNA (cDNA) was generated from the extracted RNA using specific primers and TaqMan MicroRNA Reverse Transcription Kit (Applied Biosystems, Foster City, CA, USA). The reaction was performed in a volume of 15 µL containing 5 ng of RNA, 100 mM of dNTPs, 50U/ µL of MultiScribe reverse transcriptase, 10× PCR buffer, 5× RT primer, 20 U/ µL of RNase inhibitor and nuclease free water. The amplification reaction was performed in a thermal cycler (Applied Biosystems, Foster City, CA, USA) using the following cycling profile: 16 °C for 30 min, 42 °C for 30 min and 85 °C for 5 min. The qPCR was performed using specific TaqMan primers and probes. Real-time fluorescence qPCR was performed using the Rotor-Gene 6000 thermal cycler (Corbett Research, Germany). The reaction conditions were as follows: 95 °C for 10 min, followed by 40 cycles of 95 °C for 15 s, 60 °C for 60 s. Twenty microliters of reaction mix contained: 10 µL of TaqMan Universal PCR Master Mix (Applied Biosystems, Foster City, CA, USA), 20× of Small RNA Assay, 1.33 µL of reverse transcription product and nuclease free water. *C.elegans* miRNA-39 was used as an internal control. The 2−ΔΔCt method was used to calculate the fold change in miRNA expression between the two groups. The 2−ΔCt method was used to calculate the relative expression of miRNAs in every group and the results were plotted in figures to graphically show the difference in miRNA expression between the groups [62].

## Figures and Tables

**Figure 1 ijms-21-06445-f001:**
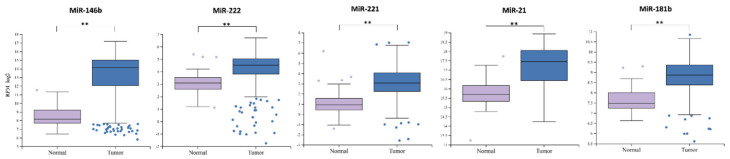
The comparison of miRNA expression between normal and cancerous tissue groups. The box squares represent the data within 25 and 75 percentiles, the line in the middle shows the median, the outliers are shown as dots. ** *p* < 0.0001. A total of 565 cases analyzed (normal *n* = 59, tumor *n* = 506).

**Figure 2 ijms-21-06445-f002:**
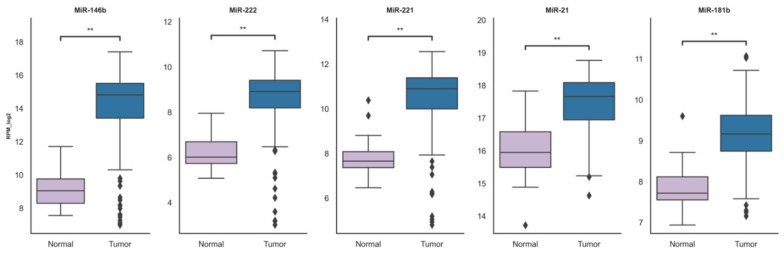
The filtered data comparison of miRNA expression between normal and cancerous tissue groups. The box squares represent the data within 25 and 75 percentiles, line in the middle shows the median, the outliers are shown as diamonds. ** *p* < 0.0001. A total of 207 cases analyzed (normal *n* = 31, tumor *n* = 186).

**Figure 3 ijms-21-06445-f003:**
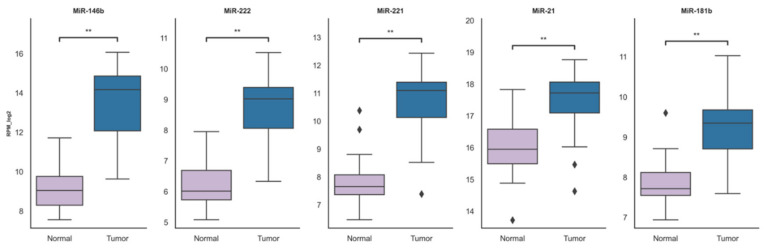
The comparison of miRNA expression between tumor and normal tissue adjacent to the tumor (normal) groups. The box squares represent the data within 25 and 75 percentiles, the line in the middle shows the median, the outliers are shown as diamonds. ** *p* < 0.0001. A total of 31 patients were analyzed (paired tumor and adjacent to the tumor (normal) 31 cases in each group).

**Figure 4 ijms-21-06445-f004:**
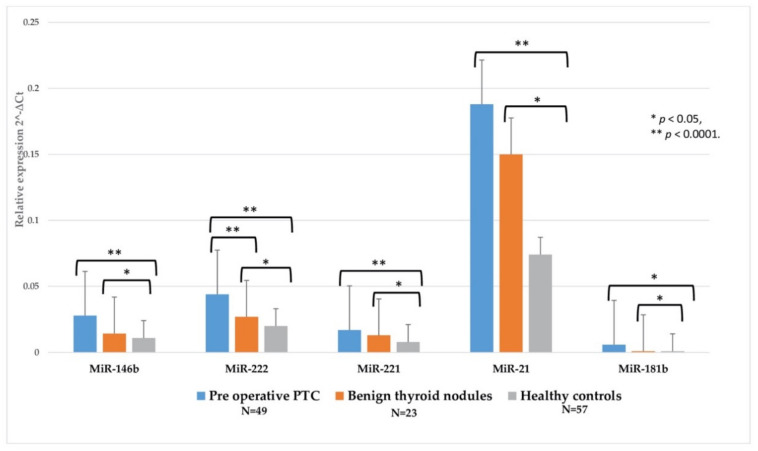
The comparison of plasma miRNA expression between papillary thyroid cancer (PTC), nodular goiter patients and healthy control groups. All data are presented as the mean ± SD. * *p* < 0.05, ** *p* < 0.0001.

**Figure 5 ijms-21-06445-f005:**
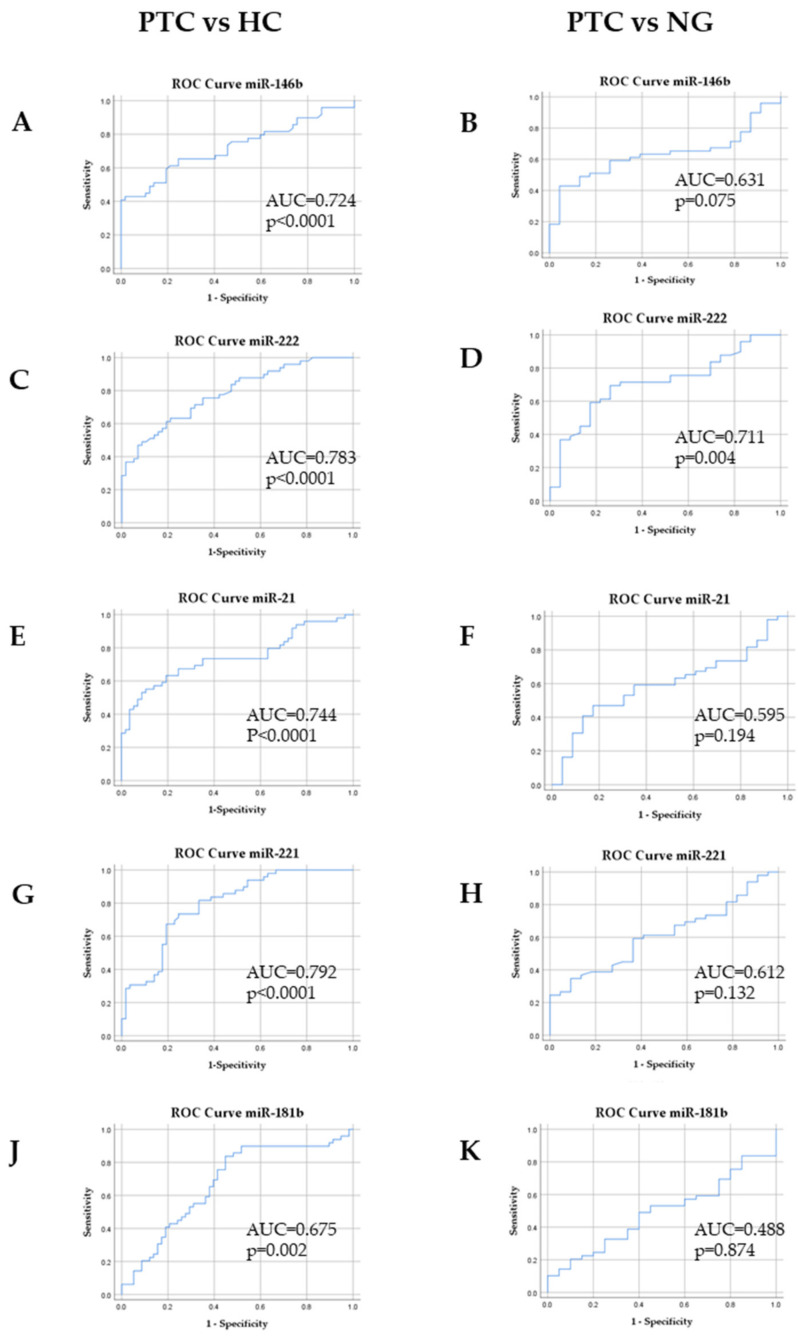
Diagnostic value of miRNAs in discriminating PTC from healthy controls (HC) and nodular goiter (NG). ROC (receiver operating characteristic) curves were used to distinguish the groups. (**A**) Plasma miR-146b in PTC vs. HC and (**B**) NG. (**C**) Plasma miR-222 in PTC vs. HC and (**D**) NG. (**E**) Plasma miR-21 in PTC vs. HC and (**F**) NG. (**G**) Plasma miR-221 in PTC vs. HC and (**H**) NG. (**J**) Plasma miR-181b in PTC vs. HC and (**K**) NG. ROC, receiver operating characteristic; AUC, area under the curve; PTC, papillary thyroid cancer; NG, nodular goiter; HC, healthy control.

**Figure 6 ijms-21-06445-f006:**
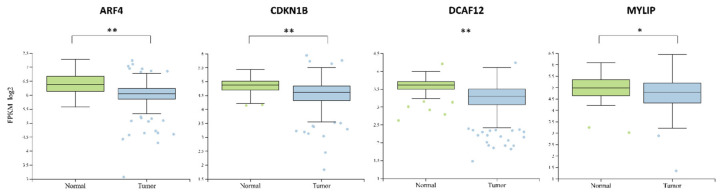
mRNA expression comparison between normal and cancerous tissue groups of the major genes modulated by miR-222. The box squares represent the data within 25 and 75 percentiles, line in the middle shows the median, the outliers are shown as dots. * *p* < 0.05, ** *p* < 0.0001. A total of 560 cases analyzed (normal *n* = 58, tumor *n* = 502).

**Table 1 ijms-21-06445-t001:** Plasma miRNA levels in papillary thyroid cancer (PTC) patients before and after total thyroidectomy.

miRNA	Relative Expression 2^−ΔCt: MEAN ± SD		Fold Change	*p* Value
	Pre-Operative PTC	Post-Operative PTC		
	(*n* = 30)	(*n* = 30)		
146b	0.029 ± 0.03	0.013 ± 0.01	1.30	0.03
21	0.18 ± 0.16	0.08 ± 0.062	1.48	0.004
222	0.04 ± 0.024	0.027 ± 0.02	1.32	0.071
221	0.018 ± 0.013	0.009 ± 0.007	1.39	0.001
181b	0.002 ± 0.005	0.0006 ± 0.001	1.23	0.038

**Table 2 ijms-21-06445-t002:** Plasma miRNA expression in papillary thyroid cancer (PTC) patients before and after hemi thyroidectomy.

miRNA	Relative expression 2^−ΔCt: MEAN ± SD		Fold Change	*p* Value
	Pre-Operative PTC	Post-Operative PTC		
	(*n* = 7)	(*n* = 7)		
146b	0.034 ± 0.032	0.028 ± 0.033	1.37	0.854
21	0.329 ± 0.231	0.175 ± 0.237	4.74	0.96
222	0.058 ± 0.038	0.049 ± 0.019	1.51	0.34
221	0.022 ± 0.016	0.008 ± 0.001	2.02	0.036
181b	0.008 ± 0.012	0.0004 ± 0.0004	10.26	0.063

**Table 3 ijms-21-06445-t003:** Clinicopathological features of papillary thyroid cancer (PTC) and relative expression of miR (-146b, -221, -21, -222, -181b) in plasma samples.

	Relative Expression 2^−ΔCt: MEAN ±SD				
PTC Clinicopathological Feature	miR-146b	miR-21	miR-221	miR-222	miR-181b
**Age**					
< 55 years (*n* = 30; 61,2%)	0.029 ± 0.03	0.191 ± 0.158	0.016 ± 0.001	0.044 ± 0.028	0.003 ± 0.007
≥ 55years (*n* = 19; 38,8%)	0.026 ± 0.021	0.183 ± 0.161	0.024 ± 0.015	0.043 ± 0.027	0.002 ± 0.006
*p*	0.967	0.853	0.361	0.864	0.967
**Gender**					
Male (*n* = 8; 16,3%)	0.026 ± 0.019	0.155 ± 0.142	0.022 ± 0.013	0.043 ± 0.018	0.008 ± 0.012
Female (*n* = 41; 83,7%)	0.028 ± 0.028	0.194 ± 0.163	0.016 ± 0.012	0.044 ± 0.029	0.002 ± 0.005
*p*	0.781	0.401	0.167	0.989	0.193
**Multifocality**					
Single (*n* = 14; 28,6%)	0.025 ± 0.027	0.164 ± 0.138	0.017 ± 0.014	0.038 ± 0.023	0.003 ± 0.007
Multiple (≥2) (*n* = 35; 71,4%)	0.036 ± 0.027	0.250 ± 0.192	0.017 ± 0.006	0.059 ± 0.032	0.001 ± 0.002
*p*	0.144	0.153	0.314	0.024	0.691
**Extrathyroidal extension**					
Yes (*n* = 35; 71,4%)	0.029 ± 0.03	0.194 ± 0.174	0.018 ± 0.012	0.045 ± 0.029	0.003 ± 0.006
No (*n* = 14; 28,6)	0.024 ± 0.02	0.172 ± 0.12	0.016 ± 0.011	0.044 ± 0.021	0.003 ± 0.006
*p*	0.842	0.833	0.400	0.765	0.535
**Lymphovascular invasion**					
Yes (*n* = 22; 44,9%)	0.03 ± 0.03	0.180 ± 0.142	0.017 ± 0.014	0.054 ± 0.031	0.004 ± 0.007
No (*n* = 27; 55,1%)	0.03 ± 0.03	0.201 ± 0.194	0.017 ± 0.011	0.041 ± 0.023	0.002 ± 0.006
*p*	0.593	0.898	0.608	0.227	0.076
**T (TNM)**					
T1a, T1b (*n* = 24; 49%)	0.033 ± 0.034	0.226 ± 0.160	0.021 ± 0.016	0.047 ± 0.026	0.005 ± 0.008
T2, T3 (*n* = 25; 51%)	0.025 ± 0.023	0.162 ± 0.159	0.015 ± 0.007	0.041 ± 0.028	0.002 ± 0.005
*p*	0.327	0.071	0.343	0.312	0.090
**Histology**					
Diffuse sclerosing variant (*n* = 12; 24,5%)	0.027 ± 0.028	0.182 ± 0.194	0.017 ± 0.008	0.044 ± 0.029	0.001 ± 0.001
Others (*n* = 37; 75,5%)	0.027 ± 0.028	0.193 ± 0.152	0.018 ± 0.014	0.043 ± 0.004	0.004 ± 0.008
*p*	0.681	0.451	0.584	0.864	0.837
**Lymph node metastases (preoperative PTC plasma samples)**					
Yes (*n* = 12; 24,5%)	0.041 ± 0.043	0.223 ± 0.2	0.015 ± 0.013	0.057 ± 0.034	0.005 ± 0.011
No (*n* = 37; 75,5%)	0.025 ± 0.015	0.174 ± 0.157	0.018 ± 0.008	0.041 ± 0.025	0.002 ± 0.005
*p*	0.276	0.555	0.618	0.358	0.618

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
