# Peer review of "Plasma-Derived miRNA-222 as a Candidate Marker for Papillary Thyroid Cancer"

_ijms, 2020, doi:10.3390/ijms21176445_

Round 1

Reviewer 1 Report

This is an interesting paper, looking at the utility of a panel of 5 miRNA's in distinguishing PTC from Healthy controls and NG. The role of miRNAs as a biomarker of cancer is a steadily increasing field with promising biomarkers coming on stream all the time. 

Minor Comments

Line 42, delete "the" after Therefore. 

Line 48, there is a hyphen in post-transcriptional

Line 52, miRNA singular

An overall review of the grammar in the paper should be conducted. There are many short bulleted sentences that reduce the flow of the paper. 

Decimal points should be used throughout the paper rather than "," this makes the data hard to read. 

Figure 1 could be improved, there are overlapping error bars and asterix which make the figure confusing. The authors should consider showing the individual data points also. More detail in the figure legend is required, spell out synonyms, include N numbers etc. 

There is no information provided as to what method of analysis was used for the PCR data, including the associated references. 

It is not clear what the utility of pre and post thyroidectomy miRNA levels are? More discussion is needed as to what purpose this informs clinically. 

It is unclear to the reviewer if the authors assessed the panel of 5 miRNAs as one singular entity in regards ROC analysis. It should be considered. 

Author Response

We have taken up the criticism and suggestions of the Reviewers. In the accompanying rebuttal letter are our point-by-point responses to each comment. In the revised manuscript, all changes have been indicated by red colored text.

Responses to the Reviewer 1 comments

Comment 1:

 Line 44, delete "the" after Therefore.

Response to comment 1:

Done.

Comment 2:

Line 50, there is a hyphen in post-transcriptional.

Response to comment 2:

Corrected.

Comment 3:

 Line 52, miRNA singular.

Response to comment 3:

Corrected.

Comment 4:

An overall review of the grammar in the paper should be conducted. There are many short bulleted sentences that reduce the flow of the paper. Decimal points should be used throughout the paper rather than "," this makes the data hard to read. 

Response to comment 4:

We decided to use language editing by MDPI.

Comment 5:

Figure 1 could be improved, there are overlapping error bars and asterix which make the figure confusing. The authors should consider showing the individual data points also. More detail in the figure legend is required, spell out synonyms, include N numbers etc. 

Response to comment 5:

Figure 1 was improved.

Comment 6:

There is no information provided as to what method of analysis was used for the PCR data, including the associated references.

Response to comment 6:

 2-ΔΔCt method was used to calculate the fold change in miRNA expression between two groups.

2-ΔCt method was used to calculate the relative expression of miRNAs in every group and the results were plotted in figures to graphically show the difference in miRNA expression between the groups.

Reference:

  • Schmittgen, T.D.; Livak, K.J. Analyzing real-time PCR data by the comparative C(T) method. Nat Protoc. 2008; 3(6), 1101-1108, doi:10.1038/nprot.2008.73

Comment 7:

It is not clear what the utility of pre and post thyroidectomy miRNA levels are? More discussion is needed as to what purpose this informs clinically.

Response to comment 7:

Revised paragraph about miRNA expression changes before and after surgery is below:

If surgery is chosen for patients with thyroid cancer<1 cm without extrathyroidal extension and cN0, the initial surgical procedure should be a thyroid lobectomy unless there are clear indications to remove the contralateral lobe. For patients with thyroid cancer >1 cm and <4 cm without extrathyroidal extension, and without clinical evidence of any lymph node metastases (cN0), the initial surgical procedure can be either a bilateral procedure (near total or total thyroidectomy) or a unilateral procedure (lobectomy). Thyroid lobectomy alone may be a sufficient initial treatment for low-risk papillary and follicular carcinomas. For patients with thyroid cancer >4 cm, or with gross extrathyroidal extension (clinical T4), or clinically apparent metastatic disease to nodes (clinical N1) or distant sites (clinical M1), the initial surgical procedure should include a near-total or total thyroidectomy (Haugen, B.R. et al. 2015). In our study, plasma levels of 5 miRNAs were measured in 37 PTC patients before and after the surgery (30 patients underwent total thyroidectomy, 7 hemi-thyroidectomy). We checked if plasma miRNA expression also decreased in the absence of the tumor. The expression levels of miR-221, miR-21, miR-181b and miR-146b were significantly lower after total thyroidectomy compared with the samples before surgery. This might show a potential of miR-221, miR-21, miR-181b, miR-146b in PTC prognosis. Only miR-21 showed a significant reduction in its plasma levels after hemi-thyroidectomy in PTC patients. This indicates that the prognostic potential of miR-221, miR-222, miR-146b, miR-181b after hemi-thyroidectomy is doubtful since no significant decline is observed 4-6 weeks after surgery. Our observations differ from other studies that investigated the levels of specific circulating miRNAs as a marker to monitor the postoperative PTC progression. Reductions of 2.7‐fold and 5.1‐fold were observed in plasma levels of miR‐222 and miR‐146b, respectively, after the surgery, though patients underwent total-thyroidectomy in this study (Lee J.C. et al. 2013).  Zhang et al. evaluated the levels of miR-222, miR-221 and miR-146b via subsequent RT-qPCR during varied postoperative periods in the same patients (Zhang Y. et al. 2017). The levels of miR-222, miR-221 and miR-146b rapidly decreased 1 month following the surgery compared with their preoperative levels in the PTC group. There was no difference in the miR-222, miR-221 and miR-146b expression levels for patients with PTC undergoing hemi-thyroidectomy or total thyroidectomy prior and after surgery (Zhang Y. et al 2017). Further studies with more subjects and longer follow up period of plasma miRNA expression after the surgery could reveal prognostic value of these miRNAs to PTC patients.

References:

  • Haugen, B.R.; Alexander, E.K.; Bible, K.C.; Doherty, G.M.; Mandel, S.J.; Nikiforov, Y.E.; Pacini, F.; Randolph, G.W.; Sawka, A.M.; Schlumberger, M.; Schuff, K.G.; Sherman, S.I.; Sosa, J.A.; Steward, D.L.; Tuttle, R.M.; Wartofsky, L. 2015 American Thyroid Association Management Guidelines for Adult Patients with Thyroid Nodules and Differentiated Thyroid Cancer: The American Thyroid Association Guidelines Task Force on Thyroid Nodules and Differentiated Thyroid Cancer. 2016, J26(1), 1-133, doi: 10.1089/thy.2015.0020.
  • Lee, J.C.; Zhao, J.T.; Clifton-Bligh, R.J.; Gill, A.; Gundara, J.S.; Ip, J.C.; Glover, A.; Sywak, M.S.; Delbridge, L.W.; Robinson, B.G.; Sidhu, S.B. MicroRNA-222 and microRNA-146b are tissue and circulating biomarkers of recurrent papillary thyroid cancer. 2013, 119(24), 4358-65, doi: 10.1002/cncr.28254.
  • Zhang, Y.; Xu, D.; Pan, J.; Yang, Z.; Chen, M,; Han, J.; Zhang, S.; Sun, L.; Qiao, H. Dynamic monitoring of circulating micrornas as a predictive biomarker for the diagnosis and recurrence of papillary thyroid carcinoma. Lett. 2017, 13, 4252–66, doi:10.3892/ol.2017.6028.

Comment 8:

It is unclear to the reviewer if the authors assessed the panel of 5 miRNAs as one singular entity in regards ROC analysis. It should be considered. 

Response to comment 8:

We calculated 5 miRNAs as one singular entity in regards ROC analysis. PTC vs NG: AUC 0.579 95% CI =0.518-0.639 (p=0.02)

PTC vs HC: AUC 0.549 95% CI =0.504-0.594 (p=0.034)

We incuded this information in the manuscript by text.

Reviewer 2 Report

In the paper entitled “Plasma-derived miRNA-222 as a candidate marker for papillary thyroid cancer” the authors report the results of the expression profiling of 5 circulating micoRNAs in individuals with PTC, nodular goiter and health controls. The expression levels of these molecules are compared to assess for differences associated with the presence of neoplastic lesions or other clinical parameters. The authors conclude that miR-222 is significantly more expressed in PTC than in individuals with benign thyroid nodules and that it is also more expressed in multifocal PTC compared to unifocal PTC. According to the values of sensitivity and specificity of a classifier based on a specific expression threshold (obtained through a ROC analysis) the authors suggest that miR-222 can be considered a candidate biomarker for molecular diagnosis of PTC.

The topic covered by the manuscript has been widely studied in the literature and despite the authors report that miR-181b and miR-21 have not been adequately investigated as plasmatic biomarker, numerous evidences in similar settings (like expression in PTC patients’ sera) have been previously presented in literature. Moreover, a number of studies and publicly available datasets with a much higher dimensionality (small RNA seq data obtained by NGS on circulating materials) are also available thus demonstrating the lack of novelty and technological adequacy of the presented study. The authors also conclude that, among the investigated miRNAs, only miR-222 is actually suitable to be considered as a candidate biomarker (as already reported in literature). Nevertheless, the proposed investigation is very focused and clear in its simple design making it worthy of consideration for readers with interest in this specific filed.

According to my opinion a number of issues must be addressed before publication.

  • In row 47 the phrase “MiRNAs are endogenous non-coding RNA molecules (19-25 nucleotides in length) identified as posttranscriptional negative regulators of gene expression by attaching to the 3’UTP of target mRNAs in the cytoplasm” is reported. It is unclear the meaning of the term UTP, does the authors mean UTR? The term UTP is not even reported in the reported reference 7.
  • The authors report the use of the 2^-∆Ct method to evaluate for relative changes in the expression of specific microRNA. This method requires the use of an endogenous control (or a spiked in control) to calculate for the ∆Ct. This control must be specified in the corresponding Materials and Methods section.
  • The authors should report the method used for the evaluation of hemolysis in plasma samples. In liquid biopsy contamination from blood cells nucleic acids may strongly influende the results of the expression measures.
  • The authors performed a very high number of different statistical test thus suggesting the possible impact of a multiple tests effect on the statistical results. May the authors comments on this or apply some method for statistical correction (Bonferroni or similar)?
  • In statistical analysis section of material and methods paragraph a number of used statistical test is reported. Nevertheless, is not clear which statistical tests (including which post hoc test) is used for the evaluations reported in figure 1. Since 3 groups of samples are compared and ANOVA test should be followed by an adequate post-hoc test (e.g. Tukey HSD test) if tests indicates statistically significant differences. Can the authors please specify?
  • The overall quality of the English language used is very poor. I strongly suggest to extensively revise and edit the document in order to meet the basic requirement for publication

Author Response

We have taken up the criticism and suggestions of the Reviewers. In the accompanying rebuttal letter are our point-by-point responses to each comment.

In the revised manuscript, all changes have been indicated by red colored text.

Responses to the Reviewer 2 comments

Comment 1: In row 47 the phrase “MiRNAs are endogenous non-coding RNA molecules (19-25 nucleotides in length) identified as posttranscriptional negative regulators of gene expression by attaching to the 3’UTP of target mRNAs in the cytoplasm” is reported. It is unclear the meaning of the term UTP, does the authors mean UTR? The term UTP is not even reported in the reported reference 7.

Response to comment 1:

Thank you for noticing a mistake. It should have been 3’UTR (3’untranslated region). We left it in a full term in a manuscript and did not use abbreviation, since it is mentioned only once. We added one more reference to be clear.

Reference:

  • Felekkis, K.; Touvana, E.; Stefanou, Ch.; Deltas, C. microRNAs: a newly described class of encoded molecules that play a role in health and disease. Hippokratia. 2010, 14(4), 236-40.

Comment 2:

The authors report the use of the 2^-∆Ct method to evaluate for relative changes in the expression of specific microRNA. This method requires the use of an endogenous control (or a spiked in control) to calculate for the ∆Ct. This control must be specified in the corresponding Materials and Methods section.

Response 2:

The Caenorhabditis elegans miRNA-39 (spike-in cel-miR-39-3p) (Qiagen, Germany) was used as a synthetic spike-in control for data normalization. An equal amount of C. elegans miR-39-3p was added to each serum sample before RNA isolation according to manufacturer's protocol.

Comment 3:

The authors should report the method used for the evaluation of hemolysis in plasma samples. In liquid biopsy contamination from blood cells nucleic acids may strongly influence the results of the expression measures.

Response 3:

The level of hemolysis in plasma samples were assessed before miRNA extraction. 100 μL of plasma was centrifuged at 1600 x g for 4 minutes at 4 °C. Oxy-hemoglobin absorbance was measured at λ = 414 nm wavelength using a NanoDrop ND1000 Spectrophotometer (ThermoFisher Scientific, USA). The procedure was repeated 3-5 times and the average optical density (OD) was calculated. Plasma samples with OD414 > 0.25 were disqualified from further analysis.

Comments 4 and 5:

  • The authors performed a very high number of different statistical tests thus suggesting the possible impact of a multiple tests effect on the statistical results. May the authors comment on this or apply some method for statistical correction (Bonferroni or similar)?
  • In statistical analysis section of material and methods paragraph a number of used statistical test is reported. Nevertheless, it is not clear which statistical tests (including which post hoc test) are used for the evaluations reported in figure 1. Since 3 groups of samples are compared and ANOVA test should be followed by an adequate post-hoc test (e.g. Tukey HSD test) if tests indicate statistically significant differences. Can the authors please specify?

Response to comments 4,5:

The description of statistical analysis was written in an concise manner. To be exact, the normality of data distribution was tested using Kolmogorov-Smirnov criteria. The association between qualitative values (age, gender) in comparative groups (PTC, HC, NG) was assessed by the Chi-square (χ2) test. Since the data distribution was not normal, Mann–Whitney U tests (miRNA expression in PTC vs NG, PTC vs HC, NG vs HC group) were used to compare two independent groups of variables. ANOVA test was not used, because the data distribution was not normal. Wilcoxon signed rank test were used to compare two dependent groups of variables (miRNA expression before and after total thyroidectomy (N30)). F (Levenes) test was used for a small sample size (miRNA expression before and after hemi-thyroidectomy (N=7)). Plasma miRNA expression relation to the clinicopathological PTC features (age (<55/≥55), gender (male/female), multifocality (yes/no), extrathyroidal extension (yes/no), lymphovascular invasion (yes/no), T (TNM) (yes/no), histology (diffuse sclerosing variant/ others), lymph node metastases (yes/no), was evaluated using ManWhitney U test since data distribution not normal.  Receiver operating characteristic (ROC) curves and the area under the ROC curve (AUC) were applied to estimate the diagnostic value of each tested miRNA for PTC. All the statistical analyses were performed using SPSS software (version 25.0, IBM, USA). P value < 0.05 was considered as statistically significant.

Comment 6:

The overall quality of the English language used is very poor. I strongly suggest to extensively revise and edit the document in order to meet the basic requirement for publication.

Response to comment 6:

We decided to use language editing by MDPI.